# Registration of Urothelial Tumours in Cancer Registries: How to Improve and Make It More Useful?

**DOI:** 10.3390/ijerph19052714

**Published:** 2022-02-25

**Authors:** Laetitia Daubisse-Marliac, Pascale Grosclaude, Marià Carulla, David Parada, Loreto Vilardell, Alberto Ameijide, Rafael Marcos-Gragera, Jaume Galceran

**Affiliations:** 1Claudius Regaud Institute, IUCT-O, Tarn Cancer Registry, CEDEX 9, 31059 Toulouse, France; pascale.grosclaude@inserm.fr; 2CERPOP, Toulouse University, Inserm UMR 1295, UPS, 31000 Toulouse, France; 3FRANCIM, Network of French Cancer Registries, 31000 Toulouse, France; 4University Hospital Center, IUCT-O, Cancer Coordination Center, CEDEX 9, 31059 Toulouse, France; 5Cancer Epidemiology and Prevention Service, Salut Sant Joan de Reus Baix Camp, Pere Virgili Health Research Institute (IISPV), Rovira i Virgili University, 43204 Reus, Catalonia, Spain; maria.carulla@salutsantjoan.cat (M.C.); alberto.ameijide@salutsantjoan.cat (A.A.); jaume.galceran@salutsantjoan.cat (J.G.); 6Anatomic Pathology Department, Salut Sant Joan de Reus Baix Camp, Pere Virgili Health Research Institute (IISPV), Rovira i Virgili University, 43204 Reus, Catalonia, Spain; david.parada@urv.cat; 7Epidemiology Unit and Girona Cancer Registry, Oncology Coordination Plan, Department of Health, Autonomous Government of Catalonia, Catalan Institute of Oncology—Girona Biomedical Research Institute, 17004 Girona, Spain; mlvilardell@iconcologia.net (L.V.); rmarcos@iconcologia.net (R.M.-G.)

**Keywords:** urothelial tumours, bladder cancer, recording, reporting, registration practices, cancer registry

## Abstract

Due to the differences in the definition, criteria of inclusion and coding of urothelial tumours (UTs), data of different cancer registries (CRs) are not comparable. The aim of this work is to study current practices of registration of UT in the European CR of the GRELL countries in order to propose new registration rules to correctly describe incidence and survival of progressive tumours like UT. A questionnaire was sent to 91 CRs to assess whether non-invasive (NI)UT, multiple UTs, UTs occurring outside or before the operating period and time between UTs are currently considered in tumour recording and reporting. All participating CRs (*n* = 42) record a NI bladder UT in sole occurrence. In case of progressive bladder UT, 98% of the CRs record at least one NIUT but 19% don’t record the invasive progression. 17% of the CRs don’t record an invasive pelvic tumour that occurs after a NI bladder UT. 19% of the CRs don’t record an invasive bladder UT that followed a NI tumour occurring outside the zone or period of time. The recording of two synchronous UTs is carried out with a grouping topography for 36% of the CRs. The same analysis conducted on the reporting of the incidence of UT also shows heterogeneity. We conclude that there is an urgent need to define clear rules for the registration of UT.

## 1. Introduction

It was estimated that in 2020, bladder cancer was the tenth most commonly diagnosed cancer globally with around 573,000 new cases diagnosed, 441,000 in men and 132,000 in women [1]. Incidence rates vary significantly among countries, being higher in those with an elevated prevalence of tobacco smoking, although other factors as infection with *Schistosoma haematobium* or exposure to natural or occupational carcinogens such as arsenic can also be a major cause in some populations [2,3]. On the other hand, when analysing geographical or time inequalities in bladder cancer statistics, variabilities in pathological diagnostic criteria or in coding, registration and reporting practices by the cancer registries (CRs) should be considered since these differences may explain at least part of such inequalities.

An old example of this was the systematic change caused by pathologists in the classification of invasive and non-invasive bladder cancer cases. Lynch et al. did a slide review of diagnostic pathologic tissue obtained from 364 bladder cancer cases diagnosed in 1983 in the state of Iowa and registered in the SEER Iowa cancer registry. A total of 162 cases had been correctly classified as invasive (*n* = 92) or non-invasive (*n* = 70) but 197 cases (45%) had been classified as invasive when they really were non-invasive, and five cases had been classified as non-invasive when they really were invasive. They concluded that the SEER program’s system tended to classify bladder cancers as invasive when, in fact, they were not [4]. Kiemeney expressed this saying that variations exists among pathologists in terms of the diagnosis and coding of low-grade tumours and that, consequently, not all Ta tumours may be registered and reported [5].

Problems can also appear in the registration process. In some registries, such as the Netherlands Cancer Registry, a T1+ tumour following a Ta tumour is registered as a new primary in order not to miss invasive tumours. In other registries, such tumours are ignored as recurrences [5].

Crow et al. analysed how CRs of the UK, Europe and USA were coding in situ, pTa and pT1 bladder cancers [6]. When comparing registration practices in the UK and the USA, the major difference was that cases of bladder carcinoma in situ and pTa transitional cell carcinoma were included in the North American cancer statistics but not in the British cancer statistics. Since 35.9% of bladder tumours registered in the UK were in the pTa or in situ categories, a significant proportion of bladder tumours were excluded from the UK incidence but included in the USA incidence values. Bladder cancer registration also varied between the CRs within mainland Europe. The differences in registration practices also affected comparisons of survival values between the UK and the USA. The exclusion of these good-prognosis lesions from the UK statistics tended to reduce UK survival rates compared with the North American values, where these lesions were included [6].

In another example, the incidence of bladder tumours may not be comparable among the Nordic countries due to varying coding practice over time concerning non-invasive tumours. From version 7.0 (December 2014) and for Finland from version 7.1, additional codes (ICD-10 D30.1, D41.1-3, D41.5-9) have been included in the definition. For Finland, the ICD-10 codes D09.0 + D41.4 were also new from version 7.0. (see: http://www-dep.iarc.fr/nordcan/English/database.asp, accessed on 1 September 2021).

Antoni et al. described several coding and registration issues that affect the computation and comparison of bladder cancer statistics. Multiple tumours occurring in the same individual could also be an issue when examining bladder cancer statistics. Since the implementation of the most recent IARC/IACR rules [7], bladder and other urothelial cancers are considered as a single entity for the purpose of counting multiple primary tumours. This implies that if a urothelial tumour appears first in the renal pelvis or the ureter, any subsequent bladder cancer in the same person may be recorded by the registry but will not be reported for statistical purposes. The same situation may arise if a non-invasive bladder tumour is recorded prior to a malignant bladder tumour in the same individual. They conclude that efforts should be made to harmonise the coding of non-invasive tumours of the bladder to improve international comparability of bladder cancer incidence, mortality and survival statistics [2].

These problems are also mentioned in the chapter on classification and coding of the different volumes of the Cancer Incidence in Five Continents (CI5) series. In the CI5 Volume XI, the authors comment that “the issue of coding non-invasive tumours (taking into account the recorded level of invasion and grade) and which to include in the tables as “cancer of the bladder” has long been a subject of debate. In CI5 Volume VI, it was decided, for the sake of geographical comparability, to exclude tumours of benign, in situ, and unspecified behaviour […]. For Volume VII, many of registries reported that they assigned the behaviour code/3 to both in situ and unspecified diagnoses, making it impossible to distinguish such cases. As a result, the editors decided to accept that non-invasive diagnoses of bladder cancer are generally considered malignant by pathologists; since Volume VII, the bladder cancer rubric (ICD-10 C67) has therefore included the in situ (ICD-10 D09.0) and unspecified (ICD-10 D41.4) categories […]. A few registries preferred not to include such cases in their dataset, even when available in the registry, for the sake of continuity over time” [8].

These problems concern not only the bladder tumours but also all tumours of the urothelium. As consequence of these problems and others, the registration of urothelial cancers raises specific issues due to the occurrence of different tumour types with a clinical course characterised by high recurrence and progression rates. It is indeed now well established that urothelial tumours (UT) often present as a continuum either between non-invasive flat tumours (pTis) and invasive malignant tumours, or between low- or high-grade papillary tumours (pTa) and invasive malignant tumours. The pathological definition of invasive tumours corresponds to invasion of the basement membrane (tumour stage equal to or greater than T1 according to the TNM classification, that is, behaviour/3). Furthermore, as some non-invasive UT are considered malignant solely on the basis of cellular anaplasia criteria, the European Network of Cancer Registries (ENCR) recommended in the mid 1990s that all UTs should be registered regardless of their behaviour (namely, invasive or not) [9]. However, this recommendation, the application of which was gradually extended to European registries, was limited to urinary bladder tumours.

Registration of non-invasive UT is now achieved, but due to the different time at which these criteria began to be applied by CRs, some countries limit the analysis of their national incidence trends only to invasive UT [8]. Faced with this situation, some CRs have organised themselves to record two evolutionary states of the same tumour in order to analyse non-invasive UT on request. However, the international rules reviewed in 2004 to standardise tumour reporting, which are still in force, only include one UT in the incidence. One of the main limitations of this rule is that it applies to all urinary tract sites, not just the bladder. Thus, although these tumours are often synchronous or metachronous multifocal, the current rules lead to reporting in incidence only the first UT diagnosed in the urinary tract, which extends over four topographic sites: renal pelvis (ICDO-3 topographic code C65), ureters (C66), bladder (C67), and urethra and multiple synchronous UT (code C68) [10]. As the second and subsequent tumours are considered to be recurrences, it is possible that some CRs may not record them, although the recommendations state that these apply to reporting rather than recording (registration), and that CRs have the option of recording more tumours than those included in the incidence.

It should be noted that these rules were enacted in order to standardise the production of indicators at a time when, on the one hand, less was known about the natural history of UT and, on the other, survival of patients with UT was shorter. Descriptive epidemiology can no longer be limited to a simple description of the incidence of tumours considered to be independent of each other. However, current recommendations do not allow a correct analysis of the incidence of tumours with a high evolutionary potential, such as UT. In addition, the expectations of urologists have changed. On the one hand, they consider that non-invasive UT are of clinical interest due to their potential for recurrence and progression despite local treatment, so they should be recorded. Furthermore, for them, unlike pathologists, the limit that defines invasion is the involvement of the bladder muscle (tumour stage equal to T2 or more), and not extension beyond the basement membrane. The management of superficial bladder tumours (Ta/Tis/T1) indeed differs from that of bladder tumours invading the bladder muscle [11]. It is therefore legitimate to expect from CRs, and for descriptive epidemiology to provide a more precise description of this clinically relevant transitional phase.

If the dynamic description of the incidence of progressive tumours now seems necessary, this must consider the fact that many CRs only cover a limited area, especially the Latin-speaking European CR, and that not all CRs have the same operational period. This makes the registration of progressive tumours more complex, and clear rules are needed to standardise the CR practices.

This study aimed to describe the current practices of recording, coding and reporting of tumours that occur in the urinary tract (IDCO-3 topographic codes C65 to C68) in Latin-language speaking European CRs by means of a comprehensive survey of the problems faced by the CR. This review allowed to verify the availability of the data and its comparability in order to develop new registration rules necessary for the production of indicators for monitoring of progressive tumours.

## 2. Materials and Methods

This study was jointly carried out by the Tarn (France), Tarragona (Spain) and Girona (Spain) CRs. The survey had two parts. The first one consisted of a questionnaire in graphic form that aimed at evaluating how 15 clinical situations are considered in the registration and reporting of tumours, and how the CR encodes these data. These situations included non-invasive UTs, multiple UTs, UTs that occur outside the registration area or before the operative period and, in the case of multiple UTs, the time between UTs (See Appendix A, Figure A1 for the filling instructions given to CR). For each situation, the questions that were asked were: “Which tumour or tumours do you record from this patient? Which tumour or tumours do you report for this patient (that is, count in incidence or submit to a database to count in the incidence)? How do you code the tumour topography, behaviour, and grade?”

The second part was designed to assess CR coding practices in two specific situations: coding tumour morphology in various situations of composite tumours (that means UT with epidermoid, glandular or neuroendocrine components) and coding the behaviour of the UT when the level of invasion is unclear in the tumour sample.

Once the methodology was developed, the questionnaires were presented during the Group of Epidemiology and Cancer Registry in Latin-speaking Countries (GRELL) meeting in 2017 in Brussels, then tested with volunteer registries and sent to 91 European CR members of the GRELL at the end of 2017. One reminder was sent and the deadline to answer the survey was 31 January 2018. All questions about the questionnaire were answered to whoever asked them. Furthermore, the authors contacted the registries by email in case of missing data or need for confirmation.

## 3. Results

A total of 42 CRs answered all the questions of the 15 situations (response rate: 46%). Figure 1 shows the participation of CR by country.

Detailed results regarding the recording, coding and reporting practices in several situations are shown in Appendix B, Table A1. The main results indicate that all CRs state that they record non-invasive bladder carcinomas (in situ and non-invasive papillary urothelial carcinomas) but some of them do not register non-invasive carcinomas from other parts of the urothelium. In the case of a non-invasive bladder tumour becoming invasive, 19% of the CRs do not record the invasive progression. When the progression has several steps (e.g., low-grade non-invasive to high-grade non-invasive and to invasive), the practices of the registries vary widely. When there are two metachronous invasive tumours in two different sites of the urinary tract, two-thirds of the CRs record both tumours and one-third only the first. The occurrence of an invasive renal pelvis UT after a non-invasive bladder UT is not recorded by 17% of the CRs. The recording of two synchronous invasive UTs at different sites (pelvis, ureter, bladder and urethra) is done with the ICDO-3 grouping code C68.9 for 36% of the CR. When the situation becomes more complex, the variability of practices in CRs increases widely. When a non-invasive bladder tumour has already been diagnosed outside the registry area or prior to registry operation, 19% of the CRs do not record the invasive progression of this tumour. When the tumour before the operating period corresponds to a specific site followed by another tumour at a different site with the same or different behaviour, the majority of the CRs record both tumours but some of them do not record any of the tumours or record only one of them.

The same analysis was performed for the reporting of UT in incidence, and it also shows high heterogeneity between CRs. For example, between one third and two thirds of CRs state that they report non-invasive bladder tumours in incidence depending on the type of non-invasive tumour.

Regarding results about morphology coding practices of composite tumours (Table 1), 81 to 90% of the CRs state they code an urothelial morphology in cases of UT with a glandular, a neuroendocrine or an epidermoid component. The coding practices for morphology vary more in the case of an almost exclusively neuroendocrine tumour with a very small urothelial component, but the majority of the CRs (57%) record a morphology code of small cell neuroendocrine carcinoma.

Finally, in relation to coding the behaviour of the tumour when the level of invasion is unclear on the tumour sample, 55% always code a non-invasive tumour, 9% always code an invasive tumour, 12% code according to grade (that is, invasive if it is high-grade, non-invasive if it is low-grade) and 24% consult a pathologist to code the behaviour.

## 4. Discussion

In this article, we present the results of a survey on urothelial tumour registration, coding and reporting criteria conducted in 91 population-based CRs from Latin-speaking European countries, of which a total 42 registries responded. The response rate varied by country. Three quarters of the French and Spanish registries answered the survey, while only one third of the Italian registries did so. The methodology, the graphic questionnaire (a rather unusual aspect) and the call for participation had been presented and explained at the 2017 GRELL annual meeting. For reasons unknown to us, few Italian cancer registries attended this meeting, unlike the French and Spanish registries. Although a completion guide was included with the questionnaire and all registries were contacted again to participate in the same way regardless of the country, it is possible that the questionnaire or the purpose of the study were less well understood by registries that had not attended the 2017 GRELL plenary meeting. Furthermore, the fact that this study was a joint French-Spanish effort may have encouraged more responses from registries in these countries, and the length of the questionnaire may have discouraged registries that were not present at the GRELL annual meeting.

These tumours have different characteristics compared to other epithelial cancers (carcinomas) in other parts of the body, and the knowledge about their biology and natural history has evolved substantially in recent years. Some of the characteristics of these cancers cause important difficulties in their exact diagnosis and, therefore, in their characterization, coding and classification, which ultimately leads to difficulties in the registration process by CRs. Among the main characteristics that cause difficulties in CRs are multicentricity, the existence of a high proportion of non-invasive tumours (carcinomas in situ and non-invasive low-grade and high-grade papillary carcinomas) with different but high risk of recurrences and progressions, and difficulties in determining the extent of invasion. On the other hand, the current international reporting criteria for these tumours are neither defined with the necessary detail nor updated.

The results of our survey show that in none of the 15 situations presented was there a unanimous response from the 42 records. In fact, there is great heterogeneity in recording, coding and reporting, especially when the situation becomes more complex and, sometimes, within the same CR; that is, in many cases the same CR does not use the same criteria in similar situations.

Most of the CR record more UT than they report. This gives them the opportunity to report them if necessary, however, the complete record of the different episodes is highly variable even in the same registry. The inclusion or not of non-invasive tumours in the incidence rates can produce very different results. In a study realized by the Tarragona Cancer Registry on the incident UTs diagnosed between 1998 and 2009 in the province of Tarragona, Catalonia, Spain, the results showed that at least 30.1% of first cases diagnosed in a patient were non-invasive low-grade, 2.6% were non-invasive high-grade, 2.0% in situ, and 61.7% invasive, while 3.5% of cases were impossible to classify. These percentages were very similar in both sexes [12]. In the United States of America, Nielsen et al. studied a cohort of 165,711 incident cases of staged primary urothelial carcinoma of the bladder diagnosed between 1988 and 2006 in the 18 SEER registries. Among these, 45% of the cases were diagnosed with Ta disease, 10% with Tis disease, 24% with T1 disease and 21% with late stage (≥T2) disease. Although the adjusted incidence rate of all stages was relatively stable, they observed that a dramatic increase in the rate of non-invasive (Ta) disease of which 77% were low-grade, and a decrease in the incidence rates for Tis and T1 cases [13]. Therefore, and according to the results of these studies, if the CR of Tarragona records and reports all first (incident) tumours (invasive and non-invasive) of the bladder, then the adjusted incidence rate will be 64% higher (17.2 per 10^5^ person-years) than if not (10.5 per 10^5^ person-years). In the SEER Program, the increase in the adjusted incidence rate will be of significantly higher.

In the study of Tarragona, the percentage of patients with recurrences without subsequent progression at five years from the first diagnosis was 34% in non-invasive low-grade cases, 27% in non-invasive high-grade cases, 14% in in situ cases and 46% in invasive cases. In addition, the percentage of cases with progression at five years from the first diagnosis were 15% in non-invasive low-grade cases, 16% in non-invasive high-grade cases and 5% in in situ cases [12]. Obviously, the proportion of cases that progress depends on the treatment. For example, the rate of progression of carcinoma in situ is influenced by the rate of cystectomies. Therefore, in the case of tumours that have progressed from non-invasive to invasive, if the Tarragona Cancer Registry only reported a urinary bladder tumour but always giving priority to the most advanced, the adjusted rate to the World standard population of non-invasive tumours would decrease from 6.8 to 5.8 while the rate of invasive tumours would increase from 10.5 to 11.5.

In another study, the Tarn Cancer Registry, France, collected the progression of all UTs (C65-C66-C67-C68, all behaviours) from 1990 to mid 1992. The number of UTs counted in the incidence was determined from several rules (whether or not there were non-invasive tumours, multiple tumours and a history). Follow-up allowed quantifying recurrences elsewhere and tumours that worsened. 342 UTs (all behaviours) were recorded in 329 patients including 11 with a history of UT. Depending on the chosen rule, the number of UTs counted ranged from 223 to 336. In 25 years of follow-up, 5% of the patients presented at least one recurrence elsewhere and 17% of non-invasive tumours became invasive over time. The clinically relevant transition to pT2 was also analysed in this study and results show that 8% of non-invasive or pT1 tumours eventually exceeded bladder muscle over time [14].

All this shows that recording (registration or not, coding and classification) and reporting (accounting or not in the statistics of incidence and survival) of UTs requires the application of criteria that should consider the combination of the following aspects: the primary site, the histology type (with especial care for neuroendocrine tumours), the grade, the extent of invasion, the multicentricity, the recurrences and the interval of time between them, the progressions and the interval of time between the first tumour and the recurrence, the difficulties in the obtaining of result of biopsies, recording or not of stage, the existence of tumours diagnosed before the registry’s period of recording, the residence of patients at the moment of diagnosis of each tumour and the standard criteria of multiplicity.

The conclusions of this study led to the setting up in 2018 of an ENCR Working Group on urinary tract tumours, involving epidemiologists from CRs, including some of the co-authors, and pathologists. The aim was to thoroughly review and update the ENCR recommendations on bladder tumours previously published in 1995. This previous version was more concerned with the rules of reporting, whereas the new rules further detail the criteria and modalities of registration considering the latest WHO classification of 2016 [15], and give recommendations to record more UTs in order to better describe the incidence and survival of these tumours in the coming years. These rules are currently being reviewed within the ENCR committee and should be published soon followed by registry training to support their implementation.

Some of these rules developed for urinary tract cancers can be used for other tumours that are known to be progressive. Due to clinical and public health interest, more and more registries record both the diagnosis of the non-invasive stage and that of the invasive stage, especially in tumours that are screened, such as breast, colorectal and cervical cancers.

We should note that anatomopathological and staging classifications are likely to further evolve in the near future in connection with biomolecular markers that allow for better typing of tumours and which are linked to the response to systemic conventional or more targeted treatments for tumours invading the bladder muscle [16,17] and to the effectiveness of the BCG therapy for pT1 high grade UT [18]. The cancer registries will therefore have to adapt to these changes by collecting more detailed prognostic data, including molecular information. This is not limited to UT but concerns all tumour locations.

## 5. Conclusions

This work summarises all the problems encountered by cancer registries in registration of multifocal, recurrent or progressive tumours, of which urothelial tumours are the perfect example. Our study shows that these problems are still present, and that there is an urgent need to harmonise recording and reporting practices in order to be able to compare incidence and survival data between territories or countries. More data also need to be collected to more accurately describe the incidence of tumours known to progress.

## Figures and Tables

**Figure 1 ijerph-19-02714-f001:**
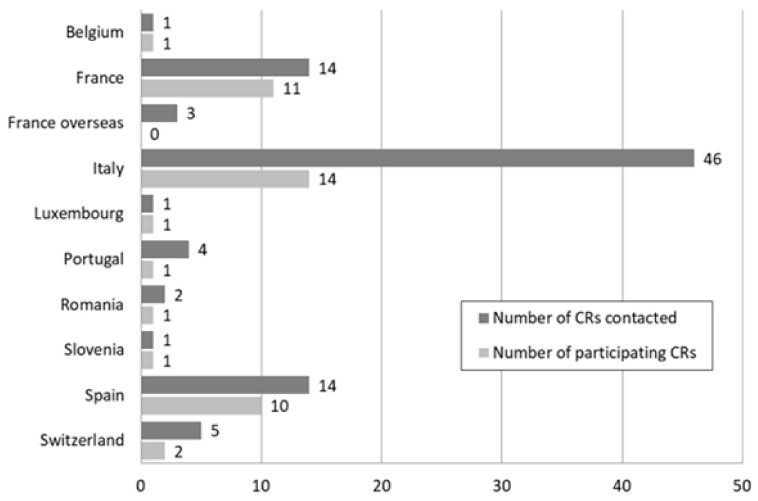
Participation of cancer registries (CRs) by country.

**Table 1 ijerph-19-02714-t001:** Morphology coding practices for composite tumours.

Type of Carcinoma	*n*	*%*
** *Urothelial carcinoma with epidermoid component* **		
**8120 Urothelial**	**38**	** *90* **
8070 Squamous	1	*2*
8575 Metaplasic	2	*5*
8120/8070	1	*2*
** *Urothelial carcinoma with adenocarcinomatous component* **		
**8120 Urothelial**	**34**	** *81* **
8140 Adenocarcinoma	4	*10*
8575 Metaplasic	2	*5*
8120/8140	1	*2*
8120/8255	1	*2*
** *Urothelial carcinoma with neuroendocrine component* **		
**8120 Urothelial**	**36**	** *86* **
8041 Small cell	3	*7*
8574 Adenocarcinoma with neuroendocrine diff.	1	*2*
8120/8041	1	*2*
8120 & 8041	1	*2*
** *Neuroendocrine carcinoma (98%) with urothelial carcinoma* **		
**8041 Small cell**	**24**	** *57* **
8045 Combined small cell	3	*7*
8120 Urothelial	4	*10*
8246 Neuroendocrine	4	*10*
8246/8041	2	*5*
8120 & 8041	2	*5*
8246/8240	1	*2*
8041/8013	1	*2*
8045/8240/8013	1	*2*

## Data Availability

The relevant data are available in the manuscript.

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
