# Peer review of "Registration of Urothelial Tumours in Cancer Registries: How to Improve and Make It More Useful?"

_ijerph, 2022, doi:10.3390/ijerph19052714_

Round 1

Reviewer 1 Report

In this manuscript the authors aimed to describe the current practices of recording, coding and reporting of urinary neoplasms. By presenting the results of a survey conducted in 91 population-based cancer registers the authors verified the availability of the data testing the comparability in order to develop a potential novel registration rules.

Overall, an underrated topic but at the same time interesting and essential for population-based analyses based on large cancer registers potentially guiding a more homogeneous criteria for the multidisciplinary managment of urothelial tumors.

Epidemiologically, a more homogeneous definition/registration criteria could represent the initial part of the multidisciplinary management of urinary bladder (or upper urinary tract) neoplasms. Particularly, this aspect should be elaborated in the Discussion paragraph.

As correctly presented in this manuscript, urothelial tumors could harbor a very different biological behaivour based on a changing geno- and phenotype. Here variant histologies or divergent differentiation were correctly referred. As a surrogate of these entities following a more specific signature, molecular subtyping require mention. All this translates into a different response - for example - to conventional chemotherapy agents and/or novel target therapies. Two recent reviews evaluated different regimens of therapy both in adjuvant (doi: 10.1016/j.euo.2021.04.004) and neoadjuvant setting (doi: 10.1016/j.ajur.2021.05.001) and may further support the discussion. Morevoer, these information could be essential to guide the treatment decision paradigm already in a specific pathologic sub-staging (pT1) since the different impact on prognosis (doi: 10.1097/JU.0000000000001422). Thus a more detailed recording, coding and reporting items could lead to a more refined clinicopath. classification.

Reviewer 2 Report

In the present article entitled “Registration of Urothelial Tumours in Cancer Registries: How to Improve and Make It More Useful”. Marliac et al summarize the issues cancer registries have when it comes to registering multifocal, recurring, and progressive tumors, like urothelial tumors. Their work shows that these issues persist and that there is a pressing need to standardize recording and reporting techniques in order to compare incidence and survival statistics across regions or nations.

Overall the manuscript is straightforward, and concise within the scope of  MDPI-International Journal of Environmental Research and public health.

However, I have minor queries which should be addressed before publishing this article:

  • The authors provide the statistics of bladder cancer from 2020. If possible, they should provide the latest statistics for the year 2022.
  • The authors describe and provide examples of heterogeneity in recording UTs. However, the authors should also discuss what can be done to address cancer registry difficulties and what advances have been made in other progressive malignancies that have resulted in improvements so that if possible those improvements can be used in UTs.

Reviewer 3 Report

This is a very interesting analysis of the coding practice within European cancer registries. It is an important topic and a potential explanation for some differences seen in epidemiological statistics. The authors used a questionnaire approach and a kind of quality ring test. Unfortunately, this was not accompanied by an Investigation regarding reasons for the differences in coding. Therefore, the study remains somwhat descriptive.

The response rate is relatively low which is almost exclusively caused by the Italian registries. Is there any explanation for this unequal distribution? Is there a relationship between coding behavior and the registry regulations in the participating countries?

Round 2

Reviewer 3 Report

The authors addressed all my comments